# Attrition Rates in Multiple Myeloma Treatment under Real World Conditions—An Analysis from the Austrian Myeloma Registry (AMR)

**DOI:** 10.3390/cancers15030962

**Published:** 2023-02-02

**Authors:** Magdalena A. Benda, Hanno Ulmer, Roman Weger, Patrick Reimann, Theresia Lang, Petra Pichler, Thomas Winder, Bernd Hartmann, Irene Strassl, Maria Theresa Krauth, Hermine Agis, Siegfried Sormann, Klaus Podar, Wolfgang Willenbacher, Ella Willenbacher

**Affiliations:** 1Internal Medicine II: Oncology, Hematology, Gastroenterology, Infectiology, Academic Teaching Hospital Feldkirch, 6800 Feldkirch, Austria; 2Private University of the Principality of Liechtenstein, 9495 Triesen, Principality of Liechtenstein; 3Institute of Medical Statistics and Informatics, Medical University of Innsbruck, 6020 Innsbruck, Austria; 4Internal Medicine V: Haematology & Oncology, Medical University of Innsbruck, 6020 Innsbruck, Austria; 5syndena GmbH, Connect to Cure, 6020 Innsbruck, Austria; 6Internal Medicine I: Hematology, Oncology, Nephrology & Endocrinology St. Pölten, Medical University of St. Pölten, 3100 St. Pölten, Austria; 7Division of Hematology with Stem Cell Transplantation, Hemostaseology and Medical Oncology, Department of Internal Medicine I, Ordensklinikum Linz, Fadingerstrasse 1 and Seilerstätte 4, 4020 Linz, Austria; 8Medical Faculty, Johannes Kepler University Linz, Altenberger Strasse 69, 4040 Linz, Austria; 9Department of Internal Medicine I, Division Hematology & Hemostaseology, Medical University Vienna, 1090 Wien, Austria; 10Internal Medicine, Division of Hematology, Medical University of Graz, 8036 Graz, Austria; 11Department of Internal Medicine II, University Hospital Krems, and Molecular Oncology and Hematology Unit, Karl Landsteiner University of Health Sciences, 3500 Krems an der Donau, Austria

**Keywords:** multiple myeloma, attrition rate, treatment patterns

## Abstract

**Simple Summary:**

Reports of attrition rates in the treatment of multiple myeloma vary widely, indicating that despite all innovations in multiple myeloma treatment, many patients do not reach their full treatment potential. In this retrospective study, the attrition rate in the Austrian Myeloma Registry (AMR) was analysed. A total of 571 patients diagnosed between January 2009 and August 2021 were included. The result of attrition in the AMR is very encouraging compared to previous data, with 22% +/− 5% per line of treatment (LoT). Attrition is higher in the elderly and lower in patients with optimal frontline treatment, including stem cell transplantation and maintenance. The importance of achieving an optimal response is highlighted, not only in terms of attrition, but also in terms of the achievable treatment-free intervals. These promising results support the putative key role of liberal universal drug access and reimbursement.

**Abstract:**

Multiple myeloma (MM) is characterized by serial relapses, necessitating the application of sequential lines of therapy (LoT). Reports on attrition rates (ARs) vary widely. The present study analysed ARs from the Austrian Myeloma Registry. Attrition was defined as being either deceased, progressive without having received another LoT, or lack of follow-up for ≥5 years. A total of 571 patients diagnosed between January 2009 and August 2021 were included (median age: 72 years; median follow-up: 50.8 months). Some 507 patients received at least one LoT. Of the total, 43.6% underwent autologous stem cell transplantation (SCT, transplant eligible = TE)) with primarily VRd (Bortezomib/Lenalidomide/Dexamethasone) given as induction (26.5%), followed by lenalidomide maintenance in 55.7% of cases. Transplant-ineligible (NTE) patients were predominantly treated with Vd (Bortezomib/Dexamethasone, 21.6%), receiving maintenance in 27.1%. A total of 37.5% received a second LoT. ARs across one to five LoTs were 16.7–27%. Frontline induction/ SCT followed by maintenance reduced ARs associated with age and achievement of deep remission in the frontline. Deep remission prolongs follow-up and time-to-next-treatment (TTNT), while high-risk-cyctogenetics negatively affected these outcomes. Our results demonstrate considerably lower ARs for MM patients within the AMR data versus other healthcare systems. Young age and the achievement of significant remissions after optimal frontline therapy resulted in particularly low ARs. These promising results support a key role for the ease of drug access and reimbursement policies in governing long-term MM patient outcomes.

## 1. Introduction

Multiple myeloma (MM) is a chronically relapsing hematological malignancy of plasma cells associated with numerous clinical complications such as anemia, renal dysfunction, and the development of bone lesions. Treatment options have improved considerably in recent years with the increasing use of novel agents and anti-myeloma antibodies [1,2]. Nevertheless, MM remains a largely incurable disease with frequent relapses and the need to regularly institute further lines of therapy (LoTs) [3].

In 2016, Yong et al. described an increase in clinical complications and symptom burden with each LoT, showing that only a minority of patients received more than five LoTs [4]. A recent US study by Fonseca et al. found an AR of 43–57% in MM patients ineligible for autologous stem cell transplant (SCT) across all LoTs and 21–37% in autologous transplant-eligible patients. [3]

The increasing improvement of treatment options and possibilities in advanced LoTs underlines the need to encourage patients to stay on therapy and fully exploit available treatment options and sequencing strategies. Patient comorbidities, age, as well as the presence of high-risk cytogenetic profiles and advanced MM stages (Revised International Staging System, R-ISS III), have been proposed to play a particular role for high ARs [3]. In contrast, liberal drug access and reimbursement policies of high-cost, modern anti-MM therapies doubtlessly reduce ARs [5]. In this respect, health care systems differ worldwide. In Austria, medicines approved by the EMA are universally reimbursed, as is the costs of hospitalisation and diagnostics ordered by specialists. Only a minor prescription fee is charged. The off-label use of therapies can be granted on individual application if sufficient medical evidence can be provided. 

To identify the causes of high ARs in order to challenge the impediments to successful treatment outcomes in the real-world-setting is of the highest interest. The present study aims to assess ARs in the Austrian health care system using the AMR database.

## 2. Materials and Methods

### 2.1. Study Design and Data Sources 

Patients documented via the AMR were included in this retrospective analysis if they received any myeloma treatment between January 2009 and August 2021. Patients with plasma cell leukaemia or smouldering myeloma are assessed in the AMR, while patients with AL amyloidoses are not included. Exclusion factors included lack of sufficient clinical data such as age, sex, and initial date of diagnosis. The starting point (JAN 2009) was chosen to exclude patients treated by outdated chemotherapy protocols alone, while the endpoint of the recruitment period was chosen to guarantee a minimal follow up. 

Due to decreasing patient numbers in advanced LoT cohorts, LoTs are recorded up to the fifth LoT. Subsequent LoTs are noted as >5 LoT without further discrimination. The analyses distinguished between patients eligible for autologous stem cell transplantation (TE) and patients not eligible for autologous transplantation (NTE). Patients who received a transplant were documented as TE. Therapeutics were stratified into proteasome inhibitors, immunomodulators, antibodies, and chemotherapy. The type of therapy is divided into doublet, triplet and quadruplet regimens. Duration of treatment (DoT) was defined as the duration between day 1 and the last day of respective LoT. Induction therapy, stem cell transplantation and maintenance therapy were considered as one LoT according to the International Myeloma Working Group (IMWG) recommendations. Time to next treatment (TTNT) was the period between the end date of the previous LoT and the start date of the subsequent LoT. Follow-up is reported and analysed as a relative indicator of the duration of the disease. The application of maintenance until toxicity/progression with lenalidomide was documented. 

Patient characteristics such as the Revised International Staging System (R-ISS), comorbidities according to the Charlson Comorbidity Index, and cytogenetic risk features were documented. For cardiovascular disease, myocardial/heart failure and peripheral/cerebrovascular disease were grouped together. Comorbidity graduation is documented for renal failure, liver disease and diabetes. 

High-risk cytogenetic features included hypodiploid karyotypes, translocation t(4;14), translocation t(14;16), deletion del(17p) and amplification 1q of copy numbers. Initial performance status in line with WHO criteria was assessed at the beginning and end of treatments, while ECOG scores were assessed at the first diagnosis. For each LoT, remission status, best response and the attrition rate were recorded. Deceased patients and patients “lost to follow-up” were included in the attrition cohort. Patients without any documentation ≥ 5 years in the registry were identified as “lost to follow-up”. Patients with progressive disease without any additional LoT were likewise added to the AR cohort. 

### 2.2. Statistical Methods

Categorical data was given with numbers (n and percentages (%) and continuous data with means and standard deviations (SDs), as well as median and interquartile range (IQRs). Statistical testing was performed using cross-tabulations and a chi-square test to assess associations between categorical variables while an analysis of variance (ANOVA) was performed for continuous data. Time-to-event data was presented using the Kaplan Meier approach together with log-rank testing. 

Finally, logistic regression analyses were used for multivariate analysis to further investigate the influence of univariate significant variables and clinically important confounders on ARs and response. Odds ratios (OR) and 95% confidence intervals (95%CI) were reported for the main outcomes, with *p* < 0.05 considered statistically significant.

## 3. Results

571 patients could be identified from the AMR database in line with the defined inclusion and exclusion criteria. For a consort diagram see Figure 1.

The median (IQRs) age (years) of patients was 72 (63–80) years, with a median follow-up time (months) of 50.8 (24–96.8) months. A total of 57.1% of the patients were male (n = 326) and 42.9% were female (n = 245). Patient characteristics (revised international staging system= R-ISS status, comorbidities and cytogenetic risk features) are depicted in Table 1; the first to third LoTs are summarized in Figure 2.

Table 1 presents standard characteristics of patients given in numbers (n) and percentage (%). Age is given in years with median and interquartile range (IQRs). Follow-up is the time from date of diagnosis until either the last contact or until August 2021, expressed in months with median and SD. Patients lost to follow-up are divided into two groups: one with recorded loss to follow-up and the other with lost to follow-up recorded during data cleaning. When missing data is identified in the database, the number of patients with recorded values is reported as “documentation performed” in numbers and percentage. Comorbidities were documented in the registry according to the Charlson Comorbidity Index without graduation, with the exception of liver disease/renal insufficiency and diabetes. For cardiovascular disease, myocardial/heart failure and peripheral/cerebrovascular disease were grouped together. High-risk cytogenetic patients include those with hypodiploid karyotype, translocation (4;14), translocation (14;16), deletion (17p) and amplification (1q) without knowledge of copy frequency. Initial performance status in line with WHO criteria was assessed at the beginning and end of treatments, while ECOG scores were assessed at first diagnosis. The performance status presented refers to the start of the first line treatment.

### 3.1. First Line Treatment (LoT 1)

Induction therapy followed by autologous stem cell transplantation (SCT) was performed in 43.2% of patients (TE patients; n = 219). The median age was 64 (56–70) years. Patients undergoing autologous SCT were significantly younger than patients not eligible for autologous SCT (78 (71–82) years; *p* < 0.001). Autologous SCT was performed in 77.5% of patients ≤ 65 years. 

The predominant induction regimens in the autologous SCT cohort were VRd (Bortezomib, Lenalidomide, Dexamethasone; 26.5%, n = 58) and VTd (Bortezomib, Thalidomide, Dexamethasone; 25.6%, n = 56). Triplet regimens were commonly used (84 %, n = 184). Maintenance therapy with Lenalidomide was prescribed to most patients (55.7%, n = 122). 

After one LoT, 68.9% of patients (n = 151) demonstrated a very good partial response (VGPR) or better response, while only 5.5% were primary progressive (n = 12, PD). Autologous SCT improved the chance of achieving VGPR or better by 3.7-fold (95%CI 4.24:3.2–6.9; *p* < 0.001). Two LoTs were performed in 38.4% (n = 84) of patients, while 16.0% fulfilled the attrition criteria (n = 35)

The median age of transplant- ineligible patients (NTE; n = 288) was 78 (71–82) years. Bortezomib/Dexamethasone (Vd; 21.6%, n = 62) and VRd (16%, n = 46) represented the predominant induction regimens among these patients. Most patients received triplet regimens (52.8%, n = 152). Continuous therapy was used in 27.1% of patients (n = 78). VGPR or better was observed in 31.9% of patients (n = 92); PR, minimal response (MR) or stable disease (SD) in 29.5% (n = 85) of patients; and PD in 12.8% (n = 37) of patients. No remission status was documented in 74 patients. (25.7%). VGPR or better was achieved more frequently in patients receiving triplet or a clinical trial regimen (*p* < 0.001 and *p* = 0.017, respectively). No difference was observed in the proportion of cytogenetic high risk patients or patients with R-ISS-III status between the TE and NTE cohorts (Appendix A). Some 37.2% (n = 107) of patients received a second LoT, while 21.9% of patients fulfilled the AR criteria (n = 63). 

The median time of treatment (DoT) was 6.6 (3.8–6.7) months, with a significant difference between TE and NTE patients (7.9 (5.3–17) months vs. 5.5 (2.8–11.9) months; *p* = 0.002). Patients receiving maintenance therapy had a longer DoT of 15.9 (10.1–32.3) months vs. 5.3 (2.9–7.5) months. No relevant differences were observed in patient characteristics such as high-risk cytogenetics, treatment patterns or response.

The median time to next treatment/treatment-free period (TTNT) was 3.8 (1.1–20.5) months with a significant difference in TE vs. NTE patients (5.3 (2.8–23.5) months vs. 2 (0.5–11.5) months; *p* = 0.009). Moreover, a significant difference was also observed in patients achieving a VGPR or better versus those who did not (10.7 (3.7–27.5) months vs 1.4 (0.4–4.7) months; *p* < 0.001). Maintenance therapy had no relevant effect on TTNT. 

Chemotherapy but no other types of treatment significantly prolonged TTNT (*p* = 0.016, 6.8 months, 1.5–23.6 months), independent of containing Anthracyclines (38.8%, n = 35), Cyclophosphamide (35.3%, n = 41) or Melphalan (24.1%, n = 28) (*p* = 0.271). The effect of chemotherapy loses significance without TE patient contrast; only 2% of patients (n = 7) completed the treatment with a quadruplet regimen, so any analysis of this new approach must wait. In patients with high-risk cytogenetics, TTNT was shorter than in patients without this feature (3.1 months, 0.9–8.3 months, vs. 4.2 months, 0.8–4.2 months; *p* = 0.033). No difference in outcome was observed in patients with a high comorbidity index, age > 70 years and R-ISS III status. 

Median follow-up was likewise positively influenced by the application of an SCT (53.8 months, 29.8–97.5 months vs. 47.4 months, 21.2–88.4 months *p* = 0.006).Patients treated with recently approved therapeutics naturally have a shorter the median follow-up as, for example, antibody treatment with 21.8 (15–27.5) months (*p* < 0.001) while “older treatments” such as chemotherapy had the opposite effect (74.8 months, 49.5–99.4 months *p* < 0.001). This can be explained by the time in which these treatments were mainly used. The use of triplet treatments positively influenced follow-up (52.8 months, 26.8–96.7 months, vs. 44.5 months, 20–87.6 months; *p* = 0.033). Patients with high cytogenetic risk tend to have significantly shorter follow-up (32.2months, 18.1–58.4 months vs 55.3months, 27.6–101.9 months; *p* < 0.001), as did patients with a high comorbidity index ≥ 4 points (29.4 months, 10.6–67 months, vs 45.4 months, 23.6–86.7; *p* = 0.004) and an ECOG score > 2 points (45.2 months, vs. 49.5 months; *p* < 0.001). An R-ISS status of llI also provided negative effects on follow up (38.5 months, SD 35 vs. 52 months, SD 44.4; *p* < 0.001).

### 3.2. Second line treatment (LoT 2)

A total of 191 patients received a second LoT; 60.7% (n = 116) were male with a median age of 72 (63–79) years. An ECOG level of >2 points at baseline was recorded in 18.8% (n = 36) of patients. Only two patients had a comorbidity status ≥ 4 points (1.1%). In 24.6% (n = 47) a high risk cytogenetic profile was documented, while an R-ISS status of III was found in 16.8% of patients (n = 32). A second LoT was applied in significantly fewer patients with a low-performance status > 2 points and an ECOG score > 2 points at baseline (18.8%, n = 36 each with *p* = 0.009 and *p* < 0.001, respectively at baseline), suggesting that fewer fragile patients are able to take up further LoTs.

TE patients (16.8% n = 32) were obviously significantly younger (63 (59–71.3) years, *p* < 0.001) than NTE patients (74 (65–81) years. Most patients received triplet regimens as induction therapy (71.9%, n = 23), VRd (28.1%, n = 9) and Rd (Lenalidomide, Dexamethason; 9.4%, n = 3) in particular. Continuous treatment was prescribed in 75% of patients (n = 24) more commonly than in NTE patients (32.3%, n = 51). More patients with R-ISS III status were encountered in the SCT Cohort than in NTE patients (n = 10, 31.3% vs. 22, 13.8%). In this cohort, VGPR or better could be achieved in 56.3% of patients (n = 18), with three patients being progressive on treatment (9.4%). The attrition criteria were met by 18.8% of patients (n = 6), with four patients deceased and two patients lost to follow up. 

NTE patients (n = 159) were commonly treated with doublet regimens (58.5%, n = 93) including Rd (37.7%, n = 60) and Vd (VD 9.4%, n = 15). Some 32.7% of patients received continuous treatment (n = 52). VGPR or better was reported in 27% of patients (n = 43) within this group, and PD in 19.5% of patients (n = 31). The response again differed significantly between TE and NTE patients (*p* = 0.015). 

The AR in the second LoT was 19.5% (n = 31), with 9.4% (n = 15) being lost to follow-up and 9.4% (n = 15) having died. Continuous treatment significantly prolonged DoT, with 22.6 (12.5–44.1) months compared to 7 (2.4–13.1) months (*p* < 0.001). 

TTNT was improved by the achievement of VGPR or better to 4.4 (1.4–16.1) months vs. 0.7 (0.2–2.5) months; *p* < 0.001). No significant statistical difference in TTNT was observed with respect to continuous treatment and high risk cytogenetics. 

Median time on follow-up was higher when continuous treatment was applied, at 82.2 (45.7–121.8) months compared to 65.9 (27.7–93.8) months with fixed duration therapy. A response of VGPR or better was also associated with a prolonged follow-up of 92.1 (54.6–113.3) months. vs. 52.8 (27.5–96-5) months for inferior responses; (*p* = 0.002). Further high-risk cytogenetics (47 (22.9–75.8) months vs. 74 (43.3–112.8) months for absence of such features; *p* < 0.001) and R-ISS status of III negatively affected time on follow-up 48.2 (16.1–80.5) months vs. 73.8 (37.8–112.9) months; *p* = 0.008.

### 3.3. Third Line Treatment (LoT 3)

Of the 85 patients receiving a third LoT, 63.5% (n = 54) were male. Their median age was 72 (64–79.5) years. Only two patients had a comorbidity score of ≥4 points (2.4%), with 23.5% of patients indicating an ECOG score >2 points at baseline (n = 20). 31.8% (n = 27) harboured high risk cytogenetics. An R-ISS of III was recorded in 18.8% (n = 16). 

The predominant treatment regimens were triplets (47.1%, n = 40), immunomodulators (24.7%, n = 21) such as Pomalidomide/Dexamethasone (18.8%, n = 16), and antibody- containing combinations (21.2%, n = 18); late and/or second autologous SCTs were performed in 11.8% of patients (n = 10). Continuous therapies were applied in 14.1% (n = 12). of patients. A subsequent LoT was administered in 37 patients (43.5%) with an AR of 23.5% (n = 20); 10 patients died, and 10 patients were lost to follow-up.

### 3.4. Fourth Line Treatment (LoT 4)

37 patients received a fourth LoT with 75.7% of patients (28n.) being male and having a median age of 72 (62–80) years. In the 4th LoT, 16.2% of patients presented an ECOG score > 2 at baseline (n = 6). No patients with a comorbidity status ≥ 4 points was recorded. High risk cytogenetics were found in 16.2% (n = 6). A total of 18.9% of patients had an R-ISS of III (n = 7). 

A total of 70.3% patients (n = 26) were treated with triplet regimens, mainly CD38-antibody containing regimens (37.8%, n = 14), most commonly in a combination of Pomalidomide/Daratumumab/Dexamethasone (13.5%, n = 5). Only 8.1% (n = 3) of patients received a first or further autologous SCT. Some 18 patients (48.6%) received a subsequent LoT. The AR was 27% (n = 10); one patient was lost to follow-up, while all others died.

### 3.5. Fifth Line Treatment (LoT 5)

A fifth LoT was received by 18 patients (61.1%, n = 11 of fourth line patients). They had a median age of 79 (66.3–81.3) years. An ECOG score >2 at baseline (n = 3) was documented in 16.7% of patients. Again, no patients with comorbidity status ≥ 4 points were observed. A total of 16.7% of patients were classified as high risk by cytogenetics (n = 3). An R-ISS of III was recorded in 5.6% (n = 1). 

Doublet and triplet regimens were commonly administered, with eight patients (44.4%) on each regimen. Treatment regimens were found to be more diverse with each subsequent LoT. The most common treatment in the 5th LoT was Dexamethasone/Daratumumab (Dd) in two patients (11.1%). Late or second autologous SCT was performed in three patients (16.6%), and maintenance therapy was initiated in 4 patients (22.2%). More than five LoTs were recorded in another 61.1% (n = 11) of patients. The AR with respect to LoT 5 was 16.7% (n = 3); two patients died, and 1 patient was lost to follow-up.

### 3.6. All LoTs

The AR across LoTs 1 to 4 was more or less constant at 22% +/− 5%, worsening only slightly with each subsequent LoT. At the fifth LoT, an AR of 16.7% (n = 3) was an outlier, which can be explained by the small number of patients pts (n = 11). The percentage of patients receiving a subsequent LoT was found to be increasing with each LoT. (Figure 3). 

The total AR across all five LoTs was 33.5% (n = 172). A total of 58.7% in this cohort were NTE (n = 101) patients. In a multivariate analysis, patients fulfilling AR criteria were significantly older (75 (64.8–82) years. vs. 70 (61–78) years; *p* = 0.011), increasing the relative risk of attrition by 86% (95% CI 1.86:1.17–2.96; *p* = 0.009). 

The use of frontline maintenance reduced the risk of AR by 50.5% (95% CI 0.5:0.33–0.74; *p* < 0.001). Autologous SCT in a frontline regimen decreased the risk of AR by 31.8% (95% CI 0.68:0.47–0.99; *p* = 0.042). Achieving VGPR or better in the first LOT further improved AR and reduced its risk by 63.5% (95%CI 0.37:0.25–0.54; *p* < 0.001). The effect is illustrated by the Kaplan-Meier curves in Figure 4.

As demonstrated by multivariate analyses, the achievement of VGPR or better was a confounder for maintenance. The use of maintenance therapy positively influenced response by increasing the chance of achieving VGPR or better by 3.6-fold (95% CI 3.61:2.37–5.51; *p* < 0.001). Age is a key factor in the decision to undergo autologous SCT and hence proved to be a confounder in multivariate analyses, as displayed in Table 2.

According to our data, the prevalence of pain, polyneuropathy and other comorbidities such as renal insufficiency did not influence ARs. Neither cytogenetic high-risk profiles nor R-ISS status (I and II vs. III) had a statistically significant impact on attrition. 

The DoT decreased continuously with a median of 6.7 (3.8–13.6) months in LoT-1 to 4 (0.5–7.3) months in LoT-5. The exception was the second LoT. This can be explained by the higher amount of patients still on treatment in the first LoT with a consecutively lower transition to the second LoT. TTNT decreased similarly with 3.8 (0.9–16.2) months in the first LoT compared to 1.5 (0.3–3.4 m) months in the fourth LoT. The course of DoT and TTNT is presented in Figure 5. 

The depth of remission decreased with each subsequent LoT, as PD was present in 9.7% of patients after the first LoT compared to 50% of patients in LoT-5. VGPR or better was achieved in 47.9% of patients in the first LoT, but only 7.1% of patients in the fifth LoT (Figure 6).

The influence on attrition across lines of therapy (LoT) 1–5 is given with an odds ratio (OR) in the upper table. The influence of sex, age over 70 years, performance status over 2 points at baseline, Revised International Staging of III (R-ISS), and high-risk cytogenetics is illustrated. The effect of stem cell transplantation (SCT) and maintenance therapy becomes less important with increasing age and the achievement of deep response (≥VGPR) in the first LoT. A possible interpretation is provided in the subsequent second table indicating factors that influence the achievement of VGPR or better in the first LOT. The categorization is performed as in the previous table. Statistically significant data are marked with *.

## 4. Discussion 

This retrospective real world-study using the AMR dataset highlights a promising AR of 16.7–27% for LoTs 1–5 compared to previously published data on the basis of a median follow up of 49.7m in both transplant-eligible and non-transplant-eligible patients [3,4]. Our results underline the importance of achieving deep remission in frontline treatments using SCT where feasible and applying Lenalidomide as maintenance treatment. 

The use of SCT shows a favorable impact on attrition with a decrease in risk to 31.8%. This is consistent with previous data [3] and recent clinical trials confirming the role of ASCT in the first LoT [6]. Whether younger age or an intensive treatment pattern improves the AR cannot be determined, as these factors are intertwined. Although the majority of patients ≤ 65 years received SCT (77.5%), TE patients were not documented as such until they had received SCT, so an immortal bias cannot be excluded. Nevertheless, SCT has been the backbone of multiple myeloma treatment for 30 years and remains so despite new agents and monoclonal antibodies [7]. The chance of achieving deeper remission and prolonged TTNT are also observed with SCT, which is in line with Yong et al. [4].

Maintenance is standard of care [8] after SCT but also administered to 27.1% of transplant ineligible patients (then usually labeled as continuous therapy), and 55.7% of transplant eligible patients, a quite positive result in comparison to previous European data [9]. Patients not receiving Lenalidomide maintenance after ASCT were usually treated before the universal acceptance and reimbursement of this standard, sometimes receiving Lenalidomid for a fixed duration of 1 or 2 years according to the former French practice. For the others, we cannot distinguish why patients did not receive treatment, so that bias cannot be completely ruled out here. Regardless, the remarkable impact of maintenance on attrition, remission, TTNT and DoT is very encouraging, and is further reenforced by Dimopoulos et al. [10]. Palumbo and Gay et al. [11] have shown the positive effects of a continuous vs. fixed dose duration of therapy; therefore, with regard to our results, the use of continuous therapies is further strengthened depending on their tolerability and toxicity. The lack of difference in TTNT otherwise lies in the very nature of maintenance, as treatment free intervals are not part of this concept. 

Depth of remission was mainly dependent on the use of maintenance and SCT (*p* < 0.001). The use of triplet regimens improved the chance of achieving VGPR or better, while doublet regimens did not have the desired effect. This effect is consistent with earlier data such as the SWOG trial S0777 [12] and was reenforced by the recent data of Jimenez-Zepeda et al. [13]. 

The predominant use of triplets in the first LoT (67.7%) is encouraging given the data from the US with the predominant use of Vd [3] and older European data with a predominant use of Bortezomib, Melphalan, and Dexamethasone as the first LoT [4,14]. The use of triplets is more dominant in transplant-eligible patients in accordance with Kumar et al. [15]. The effect of triplets on the AR has not been demonstrated, yet a greater chance of deeper remission is seen in our data, which is in agreement with Yong et al. [4]. The regimens used become more diverse with each LoT, as described by Raab et al. 2016 [14]. 

The use of ani-CD38 antibodies increased with each LoT. This is consistent with its recent approval as a first line treatment [1]. A similar observation was described by Bruno et al. [16]. The effect of antibodies on the AR seems promising, although the numbers presented here are too small to draw firm conclusions. The impact of trial participation shows the need to include more patients in clinical trials. The number of 8.9% is better than the 7% reported in European data from 2018, but further efforts for improvement should be made [17]. While the early access to innovative treatments might be reason for this phenomenon, it is possible that it may be explained by a bias due to a “healthy worker effect”. 

Depth of remission naturally had an impact on TTNT, in line with Dimopoulos [10]. DoT and TTNT across the five lines decreased comparatively. This is consistent with the results of earlier data [3,4]. These results highlights the importance of achieving deep remission in the frontline (“using the best bullet first”) and may strengthen the argument for quadruplet induction in the future. At this point we cannot comment on the effect of quadruplet induction as the follow-up of patients treated with this method is too brief and the numbers are still too small. 

The discrepant results in the fifth LoT (14 patients) and in TE patients in the third LoT (eight patients) is not seen as representative due to the small number of patients. The reported effect of chemotherapy on TTNT could be explained by the number of patients who completed the first LoT with chemotherapy (89%) compared to other therapies (61%; *p* < 0.001). However, the effect of chemotherapy loses significance without TE patients, so the use of chemotherapy-containing regimens in induction before transplantation cannot be separated from the ASCT effect.

The number of patients receiving a first-line therapy (89%) is consistent with previous data [4]. Less than half of patients receive a second line therapy (37.5%) by now and only 3.2% of patients receive a fifth line. Compared to previous data [14], this is encouraging, but the results are limited due to the short observation period, which means that a quarter of patients are still in treatment or still responding after an effective frontline regimen (see Figure 3). Regardless, the percentage of patients receiving a subsequent LoT is increasing with each LoT, possibly due to patients’ characteristics such as age and an accumulation of good biology and long-term responders to the first LoT. 

The impact of comorbidity and high risk cytogenetics on TTNT and follow-up was as expected. Follow-up decreases from the first to the fifth LoT according to disease duration. The results of follow-up in relation to maintenance, SCT and patient characteristics are interesting, as follow-up is a relative marker for disease duration. However, the results of follow-up are only representative to a limited extent due to the observation period, which is inherently limited for patients who have been diagnosed recently. The importance of evaluating cytogenetics is underscored to predict the probable course of disease. The prospect of further personalising myeloma treatment as discussed by Atrash et al. corroborates our data [18]. The low number of patients in the second LoT with a high comorbidity score, adverse ECOG-, and performance scores is in line with Fonseca et al. [3]. The missing impact of comorbidity status and high risk cytogenetics on the observed AR may be explained by a lack of statistical power given the multitude of influencing factors and the high probability of standard risk patients accumulating in the current treatment or in treatment-free remission brackets. 

A demand-based treatment with easy access to drugs might play a role in this.

Overall, our results with regard to factors impacting the attrition rate is in accordance with recent data [1]. Nevertheless, our attrition rate is quite promising in comparison. Thus, drug access and reimbursement are likely to be important factors given the cost to each LoT as well [5].

Certain limitations of this study have to be taken into account. The number of patients in the registry is limited; this limitation was accepted in the choice of observation period. The selected observation period primarily presents data on currently relevant therapies. Documentation in clinical practice is not always complete, but incomplete data have been highlighted. Nevertheless, the real word data of clinical practice are of the significant interest for the evaluation of patient management, as not all of our questions will be addressed in randomized controlled and well-designed clinical trials.

## 5. Conclusions

In summary, in utilizing the AMR database our results demonstrate considerably lower ARs for MM patients within the Austrian versus other health care systems. The low risk of attrition in our analyses was particularly associated with young age, the achievement of deep and long-lasting remissions after frontline induction/ autologous SCT, and continuous treatment. Importantly, our results emphasize a major role for a liberal drug access and reimbursement policy in governing long-term MM patient outcomes.

## Figures and Tables

**Figure 1 cancers-15-00962-f001:**
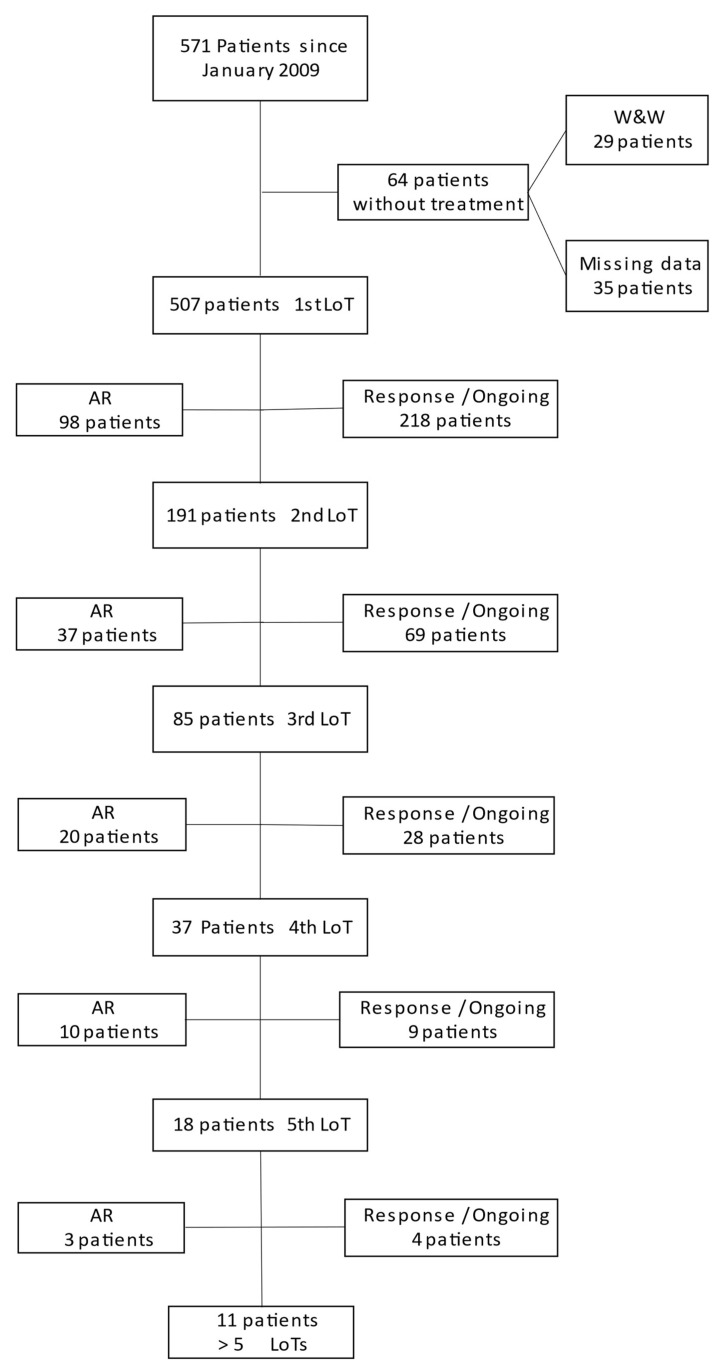
Consort Diagram. The number of patients from the beginning to the 5th line of therapy (LoT) is presented. The attrition rate (AR) is given on the left side of the diagram. Patients under treatment (ongoing) or without treatment but still in response (response) are listed on the right side for each LoT. The number of patients who have entered the subsequent LoT are displayed under each LoT.

**Figure 2 cancers-15-00962-f002:**
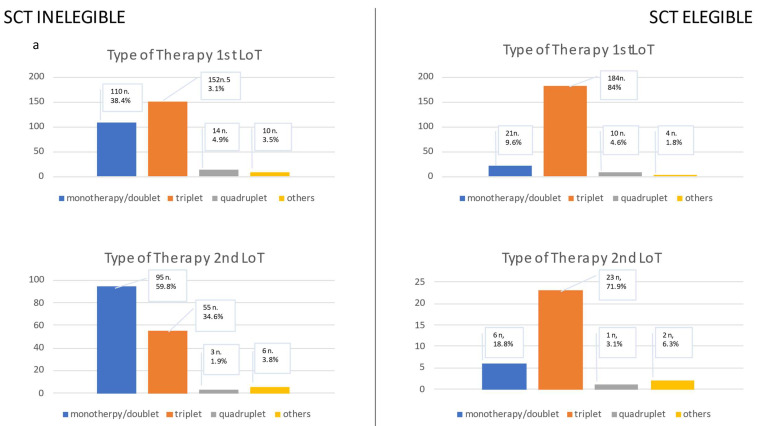
This figure presents (**a**) the type of therapy in the first line of therapy (LoT) and the second LoT. The type of therapy is divided into monotherapy and doublet, triplet, quadruplet or other regimens. The second figure (**b**) shows the type of medication in the first and second LoT. The medications are classified into the following groups: immunomodulators, proteasome inhibitors, Ps and proteasome inhibitors, chemotherapy, antibody, clinical trial, or others. In (**a**,**b**), patients are categorized into two groups: those eligible for transplantation (SCT Eligible) and those ineligible for SCT (SCT Ineligible). In the last figure (**c**), the type of medication and therapy is given for all patients in the third LoT as one group due to the limited number of patients.

**Figure 3 cancers-15-00962-f003:**
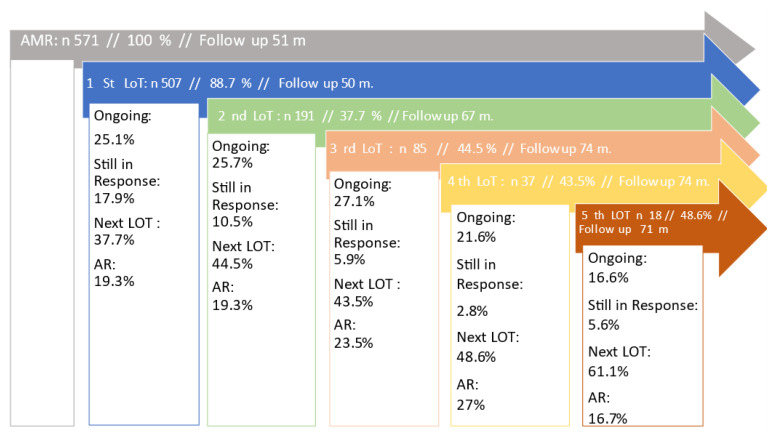
The diagram shows the number of patients in each line of therapy (LoT) from the first to the fifth LoT. Follow-up is the presenting time from date of first diagnosis to the date of last contact or data collection. The percentage given refers to the total number of the previous LOTs. The number of patients who are still receiving treatment is referred to as “ongoing” and given in percentage (%) relating to the number of patients in the present LoT. Patients who do not receive further therapy without the disease progressing are labelled as “In Response” (%). Patients who enter the next LoT are termed “next LOT” (%). “Attrition Rate” (AR %) includes deceased patients and patients lost to follow up who are either documented or detected during data cleaning.

**Figure 4 cancers-15-00962-f004:**
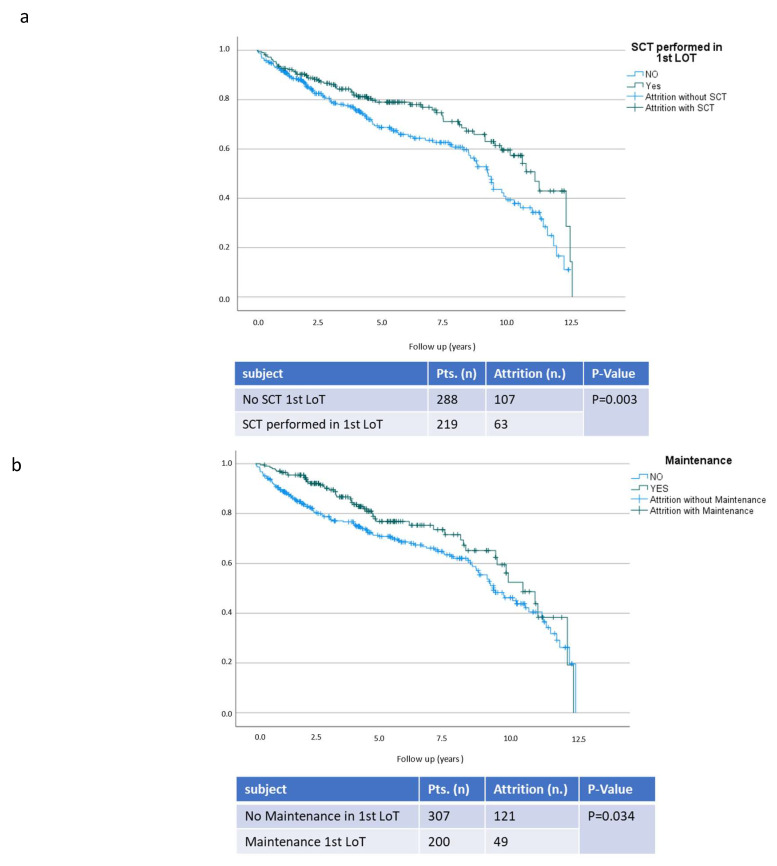
Propensity of survival without attrition in respect to stem cell transplantation, (**a**) maintenance, (**b**) and remission status (**c**). This figure presents follow-up in years and propensity of survival without attrition. On the top, patients with performed transplantation (SCT) in the first LoT are presented in comparison to transplant ineligible patients (No SCT). In the middle the difference in propensity of survival without attrition is displayed comparing patients with and without maintenance in frontline treatment. In the last diagram the propensity of survival without attrition is presented comparing patients in different remission status after the first LoT. Very good partial responses or better are presented (VGPR) in one group. Patients with partial response (PR), minimal response (MR) and stable disease (SD) are categorised in another group. Patients with progressive disease (PD) are presented separately from patients without noted remission status (n.a.).

**Figure 5 cancers-15-00962-f005:**
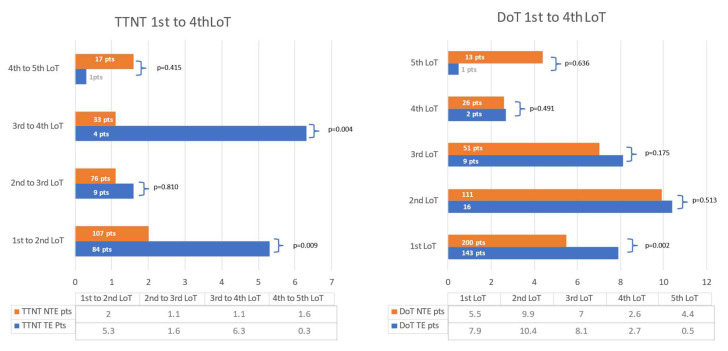
This figure presents the time to the next treatment/treatment free period (TTNT) and time of treatment (DoT) from the first to the fifth line of therapy (LoT) given as a median in months. The DoT is defined as the time from start of therapy to the end date, including stem-cell transplant and maintenance. TTNT is the time without treatment from the end date of a previous LoT to the start date of the subsequent LoT. A comparison of transplant eligible (TE) patients and transplant- ineligible (NTE) patients is displayed across the LoTs. The amount of patients analysed are given in each bar. Missing data is recorded with respect to each LoT in DoT: first LoT 37 patients, second LoT 15 patients, third LoT two patients, fourth LoT one patient, fifth LoT one patient.

**Figure 6 cancers-15-00962-f006:**
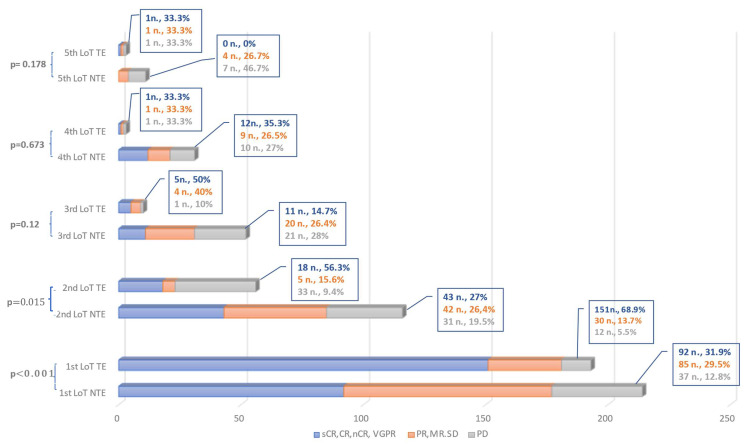
Depiction of the achieved depth of remission from the first to the fifth line of therapy (LoT). For coherence, the remission status has been categorized as very good partial response (VGPR) or better in one group and stable disease (SD), minimal response (MR) and partial response (PR) in another group. Progressive disease (PD) is listed separately. The difference between transplant-eligible (TE) and transplant-ineligible (NTE) patients is displayed for each LoT. On the right side, the number of patients and the percentage of each group is given in the same color as in the graph. The *p*-value on the left side indicates the significance in chi-square testing between TE and NTE patients.

**Table 1 cancers-15-00962-t001:** Patient characteristics.

Number of Patients			571 (n)	100(%)
		Male	326	57.1
		Female	245	42.9
Median Age			72 years	63–80 IQRs
Median Age at diagnosis			66 years	56–74 IQRs
Median Follow-up			50.8 moths	24–97 IQRs
Patient died		Total number	125	21.8
		Male	73	58.4
		Female	52	41.6
Patient “Lost of Follow-up”		Documented	29	5.1
		Not documented	16	2.8
		Male	27	60
		Female	18	40
Initial Performance status		Documentation performed	534	93.5
		Normal activity	145	27.2
		Symptoms but fully ambulatory	248	46.4
		Symptoms but in Bed <50% of the day	46	8.6
		Needs to be in bed >50%of the day	12	2.2
		Unable to get out of bed	3	0.6
ECOG Score		Documentation performed	534	93.5
		ECOG 0–2	393	73.6
		ECOG 3–4	141	26.4
Weighted index of comorbidity		Documentation performed	332	58.1
	Estimated 10- year survival	points	Number	Percentage
	98–77 (%)	0–3 points	322	56.4
	53–0 (%)	4–6 points	10	1.8
Comorbidities			Number	Percentage
		Cardiovascular	26	4.5
		Dementia	4	0.7
		COPD	3	0.5
		Connective tissue disease	0	0
		Peptic ulcer	7	1.2
		Diabetes dellitus Diabetes with end organ damage	30 2	5.2 0.35
		Moderate-severe denale disease	58	10.2
		Mild to severe liver disease	0	0
		Hemiplegia	0	0
		AIDS	0	0
		Any other malignancy within the last 5 years Lymphoma Leukaemia Metastatic solid tumor	19 4 0 0	3.3 0.7 0 0
		Neuropathy	4	0.7
		Pain	34	5.9
CRAB		Documentation performed	431	75.5
		1 CRAB Criteria	313	72.6
		2 CRAB Criteria	65	15.1
		3 CRAB Criteria	40	9.3
		4 CRAB Criteria	13	3
R-ISS III		Documentation performed	522	91.4
		R-ISS III	109	19.1
Cytogenetics		High risk Intermediate/ standard risk Not documented	124 270 177	21.7 47.3 31
Type of multiple myeloma		Documentation performed	545	95.4
		IgG	304	55.8
		IgA	103	17.9
		Free Light Chain	134	23.3
		Other (Asecretory/IgD/E/M)	4	0.7

**Table 2 cancers-15-00962-t002:** Multivariate analyses with respect to attrition and response.

Attrition Rate from LOT 1 to 5	OR	95%Confidence Interval		** *p* ** **-Value**
		Lower Value	Upper Value	
Sex	0.82	0.54	1.23	0.331
Age	1.81	1.14	2.88	0.012 *
Performance status >2 points at baseline	1.1	0.69	1.75	0.683
R-ISS III at baseline	1.31	0.78	2.19	0.309
High-Risk Cytogenetics	0.99	0.59	1.65	0.966
Maintenance 1st LoT	0.67	0.43	1.06	0.086
SCT 1st LoT	1.41	0.84	2.34	0.190
VGPR or better 1st LoT	0.36	0.23	0.57	<0.001 *
Achievement of VGPR or better 1st LOT	OR	95% confidence interval		*p*-value
		lower Value	upper Value	
Sex	0.74	0.49	1.12	0.15
Age > 70y	1.34	0.76	2.01	0.388
Performance status >2 points at baseline	1.1	0.68	1.76	0.725
R-ISS III at baseline	0.74	0.43	1.27	0.279
High-Risk Cytogenetics	1.03	0.62	1.72	0.917
Use of maintenance 1st LoT	3.61	2.37	5.51	<0.001 *
SCT 1st LoT	4.33	2.66	7.07	<0.001 *

## Data Availability

The data that support the findings of this study are available from the corresponding author, [mb], upon reasonable request.

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
