# Peer review of "Attrition Rates in Multiple Myeloma Treatment under Real World Conditions—An Analysis from the Austrian Myeloma Registry (AMR)"

_cancers, 2023, doi:10.3390/cancers15030962_

Round 1

Reviewer 1 Report

Multiple Myeloma is a disease where usually multiples lines of treatment are considered for a given patient. However, drop-outs of patients along these treatment lines are only rarely reported and frequently not captured within clinical protocols. Therefore, the data presented by the Austrian Myeloma Registry on attrition rates during Multiple Myeloma treatment are of importance and reflect standard practice in a real world setting.

Minor points:

Page 4: “Patient characteristics are depicted in table 1”. I cannot find table 1 but table 2 can be found twice. Please correct to talbe 1. 

            Issues regarding this table:

-       What exactly if the difference between “initial performance status” and “ECOG level”? In addition, should it not read “ECOG performance status”?

-       Comorbidities: what grade are they? Only grade 3 and 4? Any grade?

-       Cytogenetics: High risk as defined by? Are all remaining pts. standard risk? Or missing? It is not clear from the legend, please clarify.

-        

Please specify what is meant by maintenance therapy in first line after ASCT: fixed duration of LEN or treatment until progression/tox? Change of practice in recent years?

Please check the MS/tables carefully and correct for language and consistency. 

-       E.g.: should read cytogenetics and not cytogenetic 

-       E.g.: Last paragraph of the discussion is really difficult to read and needs numerous corrections: The number of patients in the registry is limited; this limitation was accepted when choosing the observation period in order to analyse data on the currently relevant therapies. Documentation in clinical practice is not always complete, not available data has been documented. Nevertheless real word data of clinical practice are of highest interested for evaluation of patients management as not all our questions will ever be addressed in randomized clinical trials.

Author Response

Point to point answer to the Reviewer 1’s comments:

Multiple Myeloma is a disease where usually multiples lines of treatment are considered for a given patient. However, drop-outs of patients along these treatment lines are only rarely reported and frequently not captured within clinical protocols. Therefore, the data presented by the Austrian Myeloma Registry on attrition rates during Multiple Myeloma treatment are of importance and reflect standard practice in a real world setting.

Minor points:

Page 4: “Patient characteristics are depicted in table 1”. I cannot find table 1 but table 2 can be found twice. Please correct to talbe 1. 

- Thank you very much for pointing  out this copy and paste  error, that has been  corrected.

            Issues regarding this table:

-       What exactly if the difference between “initial performance status” and “ECOG level”? In addition, should it not read “ECOG performance status”?

- A clarification was added to methods (line 118-119) and table 1: Initial Performance status in line with WHO criteria was assessed at the beginning and end of treatments while ECOG scores were assessed at first diagnosis.

-       Comorbidities: what grade are they? Only grade 3 and 4? Any grade?

- Comorbidities were documented according to the Charlson Comorbidity Index. A gradation was made for renal insufficiency/liver insufficiency and diabetes. No liver insufficiencies were documented, neither mild nor severe, and only two patients were documented to have diabetes with end-organ damage. Due to the limited number of patients with documented concomitant diseases, no grading was done and only the documented concomitant diseases were reported.

For cardiovascular diseases, myocardium/cardiac insufficiency and peripheral/cerebrovascular diseases were combined.  Under the item "tumour in the last 5 years", leukaemias and lymphomas as well as metastasised solid tumours were recorded. AIDS was not recorded. 

The comorbidities that were not recorded have now been added to table 1.

The following was added to the methods:

(lines 111-115: Patient characteristics such as the Revised International Staging System (R-ISS), comorbidities according to the Charlson Comorbidity Index and cytogenetic risk features are documented. For cardiovascular disease, myocardial/heart failure and peripheral/cerebrovascular disease were grouped together. Comorbidity graduation is documented for renal failure, liver disease and diabetes.

The following was added to Table 1:

(lines 209-212: Comorbidities were documented in the Registry according to Charlson Comorbidity Index without graduation with the exception of liver disease/ renal insufficiency and diabetes. For cardiovascular disease, myocardial/heart failure and peripheral/cerebrovascular disease were grouped together.

-       Cytogenetics: High risk as defined by? Are all remaining pts. standard risk? Or missing? It is not clear from the legend, please clarify.

Clarifications were supplemented in the table as follows:

- 124 patients had  documented high risk features , including: hypodiploid karyotype, t(4;14), t(14;16), del17p, or amp1q

- 270 patients had a standard or intermediate risk profile

- in 177 patients the cytogenetics was not documented

As well as in lines 212-214: High-risk cytogenetic patients include those with hypodiploid karyotype, translocation (4;14), translocation (14;16), deletion (17p) and amplification (1q) without knowledge of copy frequency

The following was added to the methods: lines 116-117: High-risk cytogenetic patients included those with hypodiploid karyotype, translocation (4;14), translocation (14;16), deletion (17p) and amplification (1q) agnostic of copy number.

Please specify what is meant by maintenance therapy in first line after ASCT: fixed duration of LEN or treatment until progression/tox? Change of practice in recent years?

  • The definition of maintenance therapy was included (Line 109-110: Application of maintenance until toxicity/progression with lenalidomide was documented.) Maintenance therapy in patients receiving stem cell transplantation or continuous treatment in patients not eligible for transplantation is given until progression/toxicity. In the past, a fixed duration of 2-3 years was also a standard of therapy (e.g. in France). As our data demonstrate, a relevant time to next treatment/treatment-free period is recorded in patients receiving maintenance therapy. We explain this result in the discussion: Problems with tolerability could be the reason for the lack of difference in TTNT regarding the administration of maintenance in our data.

Please check the MS/tables carefully and correct for language and consistency. 

-       E.g.: should read cytogenetics and not cytogenetic 

- Thank you very much for pointing it out. It has been corrected.

-       E.g.: Last paragraph of the discussion is really difficult to read and needs numerous corrections: The number of patients in the registry is limited; this limitation was accepted when choosing the observation period in order to analyses data on the currently relevant therapies. Documentation in clinical practice is not always complete, not available data has been documented. Nevertheless real word data of clinical practice are of highest interested for evaluation of patients management as not all our questions will ever be addressed in randomized clinical trials.:

- Thank you for noting this. The paragraph was changed as follows: “Certain limitations have to be taken into account. The number of patients in the registry is limited; this limitation was accepted in the choice of observation period. The selected observation period primarily presents data on currently relevant therapies. Documentation in clinical practice is not always complete, but incomplete data have been highlighted. Nevertheless real word data of clinical practice are of highest interested for evaluation of patients management as not all our questions will ever be addressed in randomized controlled and well-designed clinical trials.”

Reviewer 2 Report

This Austrian registry study reports attrition rates in multiple myeloma treatment in real-world data. The reporting of real-world data is important, and the study is interesting and relevant. However, there are a few comments which should be addressed before acceptance of the paper

1.       Because of the comparison to the Fonseca et al article, and because drug access and reimbursement are mentioned as a reason for low attrition rates in this study, a short explanation on drug access and reimbursement in the health system in Austria should be added to the introduction.

2.       In Introduction please clarify if you mean that “transplant patients” have undergone autologous stem cell transplant.

3.       Please clarify the inclusion criteria in methods (all patients that received myeloma treatment). Does this include patients with AL amyloidosis, plasma cell leukaemia, SMM?

4.       Give a clearer definition of how you define attrition rate in methods. How did you define “lost to follow-up”.

5.       Please clarify in Table 2 the difference between the reported performance status and ECOG?

6.       R-ISS is written R_ISS in several places, please correct

7.       The word patients is sometimes shortened to pts, sometimes pt and sometimes not shortened, this should preferably be written out in all instances. The text would be easier to read if words such as patients, years and months would not be abbreviated.

8.       Line 211: “Median follow up was likewise positively influenced by application of an SCT” this should be rephrased because this is likely caused by immortal time bias (because the patients in the SCT group would have to live long enough to get the transplant). The same applies to figure 4 (top figure) – would the patients that were intended to receive SCT (but died before this could happen) be included in the blue line? Please consider immortal time bias in all analyses with SCT and add to the interpretation.

9.       Same as in comment #8 for maintenance treatment, is there a risk of immortal time bias in these analyses?

10.   The analyses on what is associated longer follow-up are a bit confusing and do not add any relevant information. This should be omitted or explained in introduction/aims why this is performed.

11.   Line 227: “Second-line treatment had significantly fewer ps with a high-performance status >2 points and an ECOG level >2 points at baseline (18.8%, n=36 each with p=0.009 and p<0.001, respectively)” This sentence is unclear, fewer compared to what? What are you comparing here? Is baseline before first- or second-line treatment?

12.   Line 470: correct the reference.

Author Response

Point to point answer to the Reviewer 2’s comments:

This Austrian registry study reports attrition rates in multiple myeloma treatment in real-world data. The reporting of real-world data is important, and the study is interesting and relevant. However, there are a few comments which should be addressed before acceptance of the paper

  1. Because of the comparison to the Fonseca et al article, and because drug access and reimbursement are mentioned as a reason for low attrition rates in this study, a short explanation on drug access and reimbursement in the health system in Austria should be added to the introduction.

- Thank you for pointing this out. The following was added: lines 80-84: “In this respect, health care systems differ worldwide. In Austria, medicines approved by the EMA are universally reimbursed, as are the costs of hospitalisations and diagnostics ordered by specialists.  Only a minor prescription fee is charged. Off-label use of ther-apies can be granted on individual application if sufficient medical evidence can be provided. “

 In Introduction please clarify if you mean that “transplant patients” have undergone autologous stem cell transplant.

        - Thank you for pointing this out. The following was added: lines 71-73: “A recent US study by Fonseca et al. found an at AR of 43-57%  in MM patients ineligible for autologous stem cell transplant (SCT) across all  LoTs and  21-37% in autologous transplant-eligible patients”

  1. Please clarify the inclusion criteria in methods (all patients that received myeloma treatment). Does this include patients with AL amyloidosis, plasma cell leukaemia, SMM?

- Thank you for pointing this out. The following was added: lines 91-93: “Patients with plasma cell leukaemia or smouldering myeloma are assessed in the AMR, while patients with AL amyloidoses are not included.”

  1. Give a clearer definition of how you define attrition rate in methods. How did you define “lost to follow-up”.

- Thank you for pointing this out. The following was added: lines 121-122: “Patients without any documentation ≥ 5 years in the registry were identified as "lost to follow-up” “

  1. Please clarify in Table 2 the difference between the reported performance status and ECOG?

- Thank you for pointing this out. The following was added: lines 117-119 and table 1: Initial Performance status in line with WHO criteria was assessed at the beginning and end of treatment while ECOG scores were assessed at diagnose.  

  1. R-ISS is written R_ISS in several places, please correct

-Thank you for noticing. It has been corrected

  1. The word patients is sometimes shortened to pts, sometimes pt and sometimes not shortened, this should preferably be written out in all instances. The text would be easier to read if words such as patients, years and months would not be abbreviated.

-Thank you for noticing. It has been corrected

  1. Line 211: “Median follow up was likewise positively influenced by application of an SCT” this should be rephrased because this is likely caused by immortal time bias (because the patients in the SCT group would have to live long enough to get the transplant). The same applies to figure 4 (top figure) – would the patients that were intended to receive SCT (but died before this could happen) be included in the blue line? Please consider immortal time bias in all analyses with SCT and add to the interpretation.

- Only patients with documented SCT were included in the TE cohort. This was updated in the methods (lines 98-101: “The analyses distinguished between patients eligible for autologous stem cell transplantation (TE) and patients not eligible for autologous transplantation (NTE). Patients who actually received a transplant were documented as TE.”). In Figure 4 the explanation is already given (lines 427-428: “On the top patients with performed transplantation (SCT) in 1st Line of therapy (LoT) are presented in comparison to transplant ineligible patients (No SCT)”.)  The possibility of bias was added to the discussion (Line 501-503: “Although the majority of patients ≤ 65 years received SCT (77.5 %), TE patients were not documented as such until they had received SCT, so that an immortal bias cannot be excluded”)

  1. 9.       Same as in comment #8 for maintenance treatment, is there a risk of immortal time bias in these analyses?

        - The following was changed in the discussion to reflect possible bias: lines 510-514:Patients. not receiving Lenalidomide maintenance after ASCT were usually treated before the universal acceptance and reimbursement of this standard. Sometimes receiving Lenalidomide for a fixed duration of 1 or 2 years according to the former French practice. For the others we cannot distinguish why patients did not receive treatment, so that bias cannot be completely ruled out here.”

  1. The analyses on what is associated longer follow-up are a bit confusing and do not add any relevant information. This should be omitted or explained in introduction/aims why this is performed.

        - The reason for assessing follow-up is now stated in the methods (lines 108-109: “Follow-up is reported and analysed as a relative indicator of the duration of the disease”). The aim and limitation of the interpretation is further elaborated in the discussion (lines 561-565: As expected, follow-up decreases from the first to the fifth LoT according to disease duration. The results on follow-up in relation to maintenance, SCT and patient characteristics are interesting as follow-up is a relative marker for disease duration. However, the results on follow-up are only representative to a limited extent due to the observation period, which is inherently limited for patients who have been diagnosed recently)

  1. Line 227: “Second-line treatment had significantly fewer ps with a high-performance status >2 points and an ECOG level >2 points at baseline (18.8%, n=36 each with p=0.009 and p<0.001, respectively)” This sentence is unclear, fewer compared to what? What are you comparing here? Is baseline before first- or second-line treatment?

- The following was added: lines 291-294: “A second LoT was applied in significantly fewer patients with a low-performance status > 2 points and an ECOG score > 2 points at baseline (18.8%, n=36 each with p=0.009 and p<0.001, respectively at baseline), suggesting that fewer fragile patients are able to take up further LoTs”. The definition of ECOG is given in the methods as mentioned earlier.  

  1. Line 470: correct the reference.

-Thank you for noticing. It has been corrected

Reviewer 3 Report

This manuscript describes the results of an analysis of a myeloma registry in Austria. The analysis is very clinically oriented and does not contain biological/genetic data. I am surprised that the authors have submitted the manuscript to Cancers (journals such as Leukemia Lymphoma, Hematol Oncol and Eur J Haematology seem more appropriate).

The manuscript is very descriptive. I have one major problem: the text is difficult to read for readers that are not involved in myeloma treatment. The authors seem to have gone out of their way to introduce unnecessary abbreviations (that are not always explained). This starts already in the title with AMR. Examples : Vd, DoT, IQRs, SCT, VdR, VGPR etc etc.. In Fig 1 AR does not mean Attrition Rate (as it does elsewhere in the manuscript) but Attrition Criteria. TTNT is explained in Materials and Methods and in a figure legend; many readers will probably read the paper in the order Abstract > Results > Discussion (and give up).

   The text needs to be carefully revised to make it more accessible.

Author Response

Point to point answer to the Reviewer 3’s comments:

This manuscript describes the results of an analysis of a myeloma registry in Austria. The analysis is very clinically oriented and does not contain biological/genetic data. I am surprised that the authors have submitted the manuscript to Cancers (journals such as Leukemia Lymphoma, Hematol Oncol and Eur J Haematology seem more appropriate).

The manuscript is very descriptive. I have one major problem: the text is difficult to read for readers that are not involved in myeloma treatment. The authors seem to have gone out of their way to introduce unnecessary abbreviations (that are not always explained). This starts already in the title with AMR. Examples : Vd, DoT, IQRs, SCT, VdR, VGPR etc etc.. In Fig 1 AR does not mean Attrition Rate (as it does elsewhere in the manuscript) but Attrition Criteria. TTNT is explained in Materials and Methods and in a figure legend; many readers will probably read the paper in the order Abstract > Results > Discussion (and give up).

   The text needs to be carefully revised to make it more accessible.

- Thank you for the comment. The amount of patients meeting Attrition Rate criteria means Attrition Rate. The sentence has been reworded to be more understandable (lines 156: "The Attrition rate is given on the left side of the diagram”). The abbreviations have been revised to make the text more understandable.
